# Effects of Processing Conditions on the Simultaneous Extraction and Distribution of Oil and Protein from Almond Flour

**Neiva M. de Almeida** [1,2], **Fernanda F. G. Dias** [1], **Maria I. Rodrigues** [3] and **Juliana M. L. N. de Moura Bell** [1,4,*]

[1] Department of Food Science and Technology, University of California, Davis, One Shields Avenue, Davis, CA 95616, USA; neiva@cchsa.ufpb.br (N.M.d.A.); ffgdias@ucdavis.edu (F.F.G.D.)

[2] Department of Agro-industrial Management and Technology, Federal University of Paraíba, PB 58000-000 Bananeiras, Brazil

[3] Protimiza Experimental Design, 13000-000 Campinas, Brazil; protimiza@protimiza.com.br

[4] Department of Biological and Agricultural Engineering, University of California, Davis, One Shields Avenue, Davis, CA 95616, USA

[*] Correspondence: demourabell@ucdavis.edu or jdemourabell@ucdavis.edu; Tel.: +1-530-752-5007

**Abstract:** The enzyme-assisted aqueous extraction process (EAEP) is an environmentally friendly strategy that simultaneously extracts oil and protein from several food matrices. The aim of this study was to investigate the effects of pH (6.5–9.5), temperature (45–55 °C), solids-to-liquid ratio (SLR) (1:12–1:8), and amount of enzyme (0.5–1.0%) on the extraction and separation of oil and protein from almond flour using a fractional factorial design. Oil and protein extraction yields from 61 to 75% and 64 to 79% were achieved, respectively. Experimental conditions resulting in higher extractability were subsequently replicated for validation of the observed effects. Oil and protein extraction yields of 75 and 72% were achieved under optimized extraction conditions (pH 9.0, 50 °C, 1:10 SLR, 0.5% (*w/w*) of enzyme, 60 min). Although the use of enzyme during the extraction did not lead to significant increase in extraction yields, it did impact the extracted protein functionality. The use of enzyme and alkaline pH (9.0) during the extraction resulted in the production of more soluble peptides at low pH (5.0), highlighting possible uses of the EAEP skim protein in food applications involving acidic pH. The implications of the use of enzyme during the extraction regarding the de-emulsification of the EAEP cream warrant further investigation.

**Keywords:** enzyme-assisted aqueous extraction process; oil extraction; protein extraction; almond flour; solubility

## 1. Introduction

The intrinsic composition of almonds, which has long been recognized as nutritious and healthy, has enabled its use in a wide range of applications such as nutritive snacks, bakery, and confectionery ingredients, and as a feedstock to produce almond milk, yogurt, and oil [1–3]. The California almond industry accounts for approximately 80% of the world's almond supply and nearly 100% of the almond supply in the United States, with an estimated production of 2.2 billion pounds for 2017 [2,4].

Major downstream processing challenges in the fractionation of oil-bearing materials into added-value compounds (oil, protein, and carbohydrates) are usually associated with the simultaneous attainment of high extraction yields and adequate product functionality [5,6]. In general, oil-bearing materials can be extracted by the use of screw pressing or by solvent extraction. While the former leads to the production of specialty oils, extraction yields (70–90%) are not as high as the ones obtained by

solvent extraction (usually > 95%) [7,8]. Although higher extraction yields are achieved by solvent extraction, the use of flammable solvents such as hexane has raised several safety and environmental concerns [9], which has prompted the search for the development and use of more environmentally friendly extraction strategies [10–12].

The enzyme-assisted aqueous extraction process (EAEP) is an environmentally friendly strategy that enables the simultaneous extraction of oil, protein, and carbohydrates from many food materials. This water- and enzyme-based extraction technique enables the fractionation of food materials into fractions (oil-, protein-, and fiber-rich fractions) that can be further converted into food, feed, and fuel [5,10–12]. While the use of the EAEP has been demonstrated for several oil-bearing materials, its development and application for different almond fractions (flour, press cake, almond paste) has been limited to a few applications, with oil extractability being the main focus [13,14]. To the best of our knowledge, current research has not yet investigated the effects of the EAEP on the simultaneous extraction of oil and protein from almond flour, neither described the oil and protein distribution among the fractions generated by the EAEP (free oil, cream-oil-rich fraction, skim-protein-rich fraction, and insoluble-fiber-rich fraction), which might influence the final protein functionality thus determining subsequent industrial applications of the extracted protein.

Extraction parameters such as the amount and type of enzyme, pH, temperature, and reaction time have a significant impact on oil and protein extractability. In addition, they also affect the distribution of the extracted oil among the fractions generated by the EAEP. While oil and proteins have a clear and distinct application when present in its own form, the presence of residual oil in the protein-rich fraction (skim) has shown to reduce the extracted protein solubility [5]. In that view, it becomes necessary to understand the effects of extraction conditions on oil and protein extractability as well as on the distribution and functionality of the extracted compounds. The main goal of this study was to evaluate how processing variables influence overall extractability and distribution of oil and protein from the almond flour as well as the extracted protein solubility. The specific objectives of this work were to: (i) evaluate the effectiveness of two proteases regarding the simultaneous extraction of oil and protein from almond flour; (ii) evaluate the effects of extraction parameters (pH, amount of enzyme, temperature, and solids-to-liquid ratio) on extraction and separation of oil and protein from almond flour using a fractional factorial design followed by subsequent experimental validation of best extraction conditions; (iii) determine the solubility and electrophoretic peptide profile of the protein obtained under the experimental validation of best extraction conditions.

## 2. Materials and Methods

### 2.1. Material

Almond flour was kindly provided by Blue Diamond Growers (Sacramento, CA, USA). In general, the industrial production of almond flour includes the blending of blanched whole almonds (*Prunus amygdalus*) and blanched screenings, which are further milled and subjected to a screening step to produce flours with different granulometry (extra-fine to fine). The sample used in this study was produced by blending in equal amounts of whole almonds and blanched screenings of a blend of Californian varieties that were subsequently sieved through a US#12 mesh (1.70 mm sieve size), with a minimum recovery of 85%. Oil, protein, and moisture content (%) of the almond flour were 42.63 ± 0.62, 21.73 ± 0.62, and 5.37 ± 0.06, respectively. Starting material characterization was performed according to methods described in Section 2.4.

### 2.2. Enzyme Screening for the Enzyme-assisted Extraction Process (EAEP) of Almond Flour

Two commercial endoproteases (BIO-CAT, Troy, VA, USA) were tested for the EAEP: (i) Neutral Protease 2 million (NP2M), a bacterial neutral protease from *Bacillus subtilis* with enzyme activity at pH ranging from 5.5 to 9.0 and temperature from 30 to 70 °C, and (ii) Neutral Protease L (NPL), a bacterial

neutral protease from *Bacillus amyloliquefaciens* with enzyme activity at pH ranging from 5.5 to 9.0 and temperature from 35 to 80 °C.

Extraction conditions for the enzyme screening were chosen according to the manufacturer's recommendations and available data in the literature [12,15]. A process flow diagram for the enzyme screening of the EAEP of almond flour is shown in Figure 1. During the enzyme screening, 70 g of almond flour were dispersed into water to achieve a solids-to-liquid ratio (SLR) of 1:10 in a 2 L beaker. The slurry pH was adjusted to 9.0 before adding 0.5% of protease (*w/w*; weight of enzyme/weight of almond flour) and the slurry was stirred at 120 rpm for 1.5 h at 50 °C. Following the extraction, the slurry was centrifuged at 3000 × g for 30 min to remove the insoluble fraction from the liquid fraction (Figure 1).

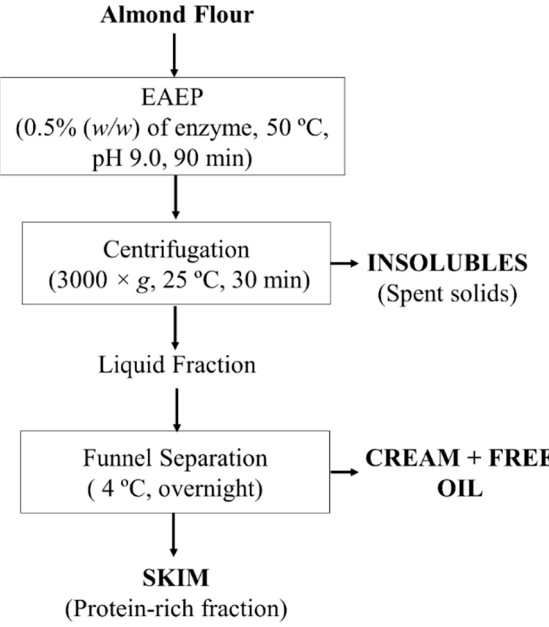

**Figure 1.** Process flow diagram for the enzyme-assisted aqueous extraction process from almond flour. Enzyme screening extraction conditions (pH 9.0, solid-to-liquid-ratio (SLR) 1:10, 0.5% of protease (*w/w*), 1.5 h at 50 °C).

The liquid phase was placed in a separatory funnel and allowed to settle overnight at 4 °C. After settling, the liquid phase was separated into three fractions (skim-protein-rich fraction, cream-oil-rich fraction, and free oil). Insoluble, skim and cream fractions were stored at −20 °C for subsequent analysis. Extractions were performed in triplicate for each enzyme.

Skim, cream, insoluble, and starting material were evaluated for oil, protein, and solids content. Total oil and protein extraction yields, which represent all oil and protein extracted from the almond flour, and the distribution of the extracted compounds among the fractions generated by the EAEP were calculated as described by Souza et al. [16]. Total oil extraction yield (TOE), oil distribution in the fractions (free oil yield, oil yield in the cream, oil yield in the skim, and oil yield in the insoluble), total protein extraction yield (TPE), and protein distribution in the fractions (protein yield in the cream, protein yield in the skim, and protein yield in the insoluble) were determined according to Equations (1)–(4), respectively.

$$TOE \ (\%) = \left[ 100 - \left( \frac{Oil \ (g) \ in \ the \ insoluble \ fraction}{Oil \ (g) \ in \ the \ almond \ flour} \right) \right] \times 100 \tag{1}$$

$$Oil \ distribution \ in \ each \ fraction \ (\%) = \left( \frac{Oil \ (g) \ in \ each \ fraction*}{Oil \ (g) \ in \ the \ almond \ flour} \right) \times 100 \tag{2}$$

$$TPE\ (\%) = \left[100 - \left(\frac{Protein\ (g)\ in\ the\ insoluble\ fraction}{Protein\ (g)\ in\ the\ almond\ flour}\right)\right] \times 100 \tag{3}$$

$$Protein\ distribution\ in\ each\ fraction\ (\%) = \left(\frac{Protein\ (g)\ in\ each\ fraction*}{Protein\ (g)\ in\ the\ almond\ flour}\right) \times 100 \tag{4}$$

where fraction * corresponds to cream, skim or insoluble.

The enzyme resulting in higher oil and protein extraction yields and/or better distribution of the extracted compounds (i.e., higher amount of oil in the cream or in the form of free oil, which would lead to a skim fraction with a lower oil content) was selected to further evaluate how extraction conditions such as pH, amount of enzyme, temperature, and SLR influence overall extractability and distribution of oil and protein from almond flour in the EAEP.

## 2.3. Effects of Extraction Variables on the Enzyme-assisted Aqueous Extraction Process (EAEP) of Almond Flour

Understanding the effects of processing variables is of key importance when aiming to reduce the number of experiments and improve process efficiency and productivity [17]. An experimental design was used to evaluate how operational parameters (pH, amount of enzyme, temperature, and SLR ratio) can affect the extraction and separation of oil and protein from almond flour. Because of the high number of variables, a fractional factorial design ($2^{4-1}$, plus three central points, totaling 11 experiments) was used for the evaluation of the effects of the variables on the responses.

The effects of the independent variables (pH: 6.5–9.5, temperature: 45–55 °C, SLR: 1:12–1:8, and amount of enzyme: 0.5–1.0%) were evaluated on the extraction and distribution of oil and protein from almond flour (Table 1). The variable levels used in the experimental design were selected based on preliminary tests as well as on existing literature data [15,18]. Dependent variables (i.e., evaluated response) were total oil extraction yield (TOE) (%), oil yield in the cream (OYC) (%), oil yield in the skim (OYS) (%), oil yield in the insoluble (OYI) (%), free oil yield (FOY) (%), total protein extraction yield (TPE) (%), protein yield in the cream (PYC) (%), protein yield in the skim (PYS) (%) and protein yield in the insoluble (PYI) (%).

**Table 1.** Fractional factorial design matrix ($2^{4-1}$, with four independent variables and three repetitions in the central point) for oil and protein extraction from almond flour.

| Experiment # | pH ($x_1$) | | Temperature (°C) ($x_2$) | | Solids-to-Liquid Ratio (SLR) ($x_3$) | | Enzyme (%) ($x_4$) | |
|---|---|---|---|---|---|---|---|---|
| | Coded Value | Real Value | Coded Value | Real Value | Coded Value | Real Value | Coded Value | Real Value |
| 1 | −1 | 6.5 | −1 | 45 | −1 | 1:12 | −1 | 0.50 |
| 2 | 1 | 9.5 | −1 | 45 | −1 | 1:12 | 1 | 1.00 |
| 3 | −1 | 6.5 | 1 | 55 | −1 | 1:12 | 1 | 1.00 |
| 4 | 1 | 9.5 | 1 | 55 | −1 | 1:12 | −1 | 0.50 |
| 5 | −1 | 6.5 | −1 | 45 | 1 | 1:8 | 1 | 1.00 |
| 6 | 1 | 9.5 | −1 | 45 | 1 | 1:8 | −1 | 0.50 |
| 7 | −1 | 6.5 | 1 | 55 | 1 | 1:8 | −1 | 0.50 |
| 8 | 1 | 9.5 | 1 | 55 | 1 | 1:8 | 1 | 1.00 |
| 9 | 0 | 8.0 | 0 | 50 | 0 | 1:10 | 0 | 0.75 |
| 10 | 0 | 8.0 | 0 | 50 | 0 | 1:10 | 0 | 0.75 |
| 11 | 0 | 8.0 | 0 | 50 | 0 | 1:10 | 0 | 0.75 |

Initial evaluation of the extraction kinetics was performed at the central point (pH 8.0, 50 °C, 1:10 SLR and 0.75% (*w/w*) of enzyme), with time points ranging from 1 to 5 h, to assess the best extraction time range to be evaluated during the fractional design. Based on the extraction kinetics data at the central point, which demonstrated no significant increase in oil and protein extractability when reaction time increased from 1 to 5 h, extractions were performed for 40 min (to verify the possibility of

achieving high extraction yields at reaction time < 1 h) and 3 h (evaluated as a conservative approach) for each experimental condition established by the experimental design.

Best experimental conditions identified by the factorial design were validated in triplicate for the enzyme assisted extraction process (EAEP) along with the aqueous extraction process (AEP) which was used as a control (same conditions without the use of enzyme). Approximately 50 g of the AEP and EAEP skim fractions obtained in the experimental validation of the best extraction conditions were freeze-dried on a benchtop freeze dryer VirTis-BenchTop™ "K" Series (SP-Scientific, Gardiner, NY, USA) and stored at –20 °C for subsequent functionality assays.

*2.4. Proximate Analysis*

Oil, protein, and dry matter contents were determined in all fractions: skim, insoluble, cream, and almond flour (starting material). Oil content was determined by using the acid hydrolysis Mojonnier method (AOAC method 922.06) [19], protein content by using the Dumas combustion method and a nitrogen conversion factor of 5.18 (Vario Max Cube, Elementar Analysensysteme GmbH, Langenselbold, Germany), and total solids (dry matter) by weighing after drying the samples in a vacuum oven (AACC Method 44–40) [20]. Extraction yields were expressed as percentages of each component in each fraction relative to the initial amounts in the almond flour according to Equations (1) to (4). All analyses were conducted in duplicate for each sample and a mass balance was provided for all extracted compounds.

*2.5. Degree of Hydrolysis*

The degree of hydrolysis (DH) of AEP and EAEP skim fractions was evaluated by the o-phthaldialdehyde (OPA) method described by Nielsen et al. [21]. Briefly, 400 μL of skim was added to 3 mL of freshly prepared OPA reagent. The mixture was vortexed and let stand for 2 min at room temperature before measuring the absorbance at 340 nm. A 0.9516 meqv/L L-serine solution was used as standard. A blank solution was prepared with distilled water instead of sample and used as the reaction control. Protein quantification (%) was determined by the Dumas combustion method (nitrogen conversion factor of 5.18) and the DH was determined as described below (Equation (5)):

$$DH\ (\%) = \frac{h}{h_{tot}} \times 100 \tag{5}$$

where $h$ is the number of hydrolyzed bonds and $h_{tot}$ is the total number of peptide bonds per protein equivalent (7.58 eq/kg for almond protein [22]).

*2.6. Low Molecular Weight (MW) Polypeptide Profile Characterization of AEP and EAEP Skim Proteins by SDS-PAGE*

SDS-PAGE was used to determine the low MW protein profile of AEP and EAEP skims obtained under experimental validation of best extraction conditions. Skim fractions were mixed with (1:1, *v/v*) Laemmli solution, vortexed and placed in a water bath (95 °C, 5 min), as described by Laemmli [23]. A Tris-HCl buffer (25 mM Tris, 192 mM glycine, 0.1% SDS, pH 8.3) (Bio Rad, Hercules, CA, USA) was used as a running buffer. Electrophoretic separation of proteins was performed by loading 30 μg of protein/well onto a precast 12% acrylamide gel (CriterionTM TGX Precast Gels, Bio Rad, Hercules, CA, USA). Electrophoretic separation was carried out at 200 V at room temperature for 1 h. A low MW range SDS-PAGE standard (14.4–97.4 kDa) (Bio Rad, Hercules, CA, USA) was used. Relative quantification and polypeptide distribution were performed using a Gel DOCTM EZ Imager system and Image Lab software (Bio-Rad, Hercules, CA, USA).

*2.7. AEP and EAEP Skim Protein Solubility*

Protein solubility of the AEP and EAEP freeze-dried skim fractions obtained by experimental validation of best extraction conditions was determined as described by Rickert et al. [24] with few

modifications. A total of 10 mL of a 1% (*w/v*) skim solution diluted in DI water was placed in a 30 mL beaker and the pH of the protein solution was adjusted to 5.0 and 9.0 by adding 1 M HCl or 1 M NaOH solution. The dispersions were stirred for 1 h at room temperature and then centrifuged at 10,000 × *g* at 20 °C for 10 min. The protein content of the supernatant was measured using the Biuret method. The protein quantification was performed using a standard curve ranging from 0 to 18 mg/mL of bovine serum albumin ($R^2$ = 0.997). The total protein content was measured after solubilizing the samples in a 1 M NaOH solution [25]. All samples were analyzed in triplicate. The solubility (%) was determined as follows (Equation (6)):

$$Solubility\ (\%)\ \frac{Protein\ in\ the\ supernatant\ (mg\ /mL)}{Total\ protein\ (mg\ /mL)} \times 100 \tag{6}$$

### 2.8. Statistical Analyses

Replicates of each measurement were analyzed by using Analysis of Variance (ANOVA) with generalized linear models from the SAS system (version 9.4, SAS Institute Inc., Cary, NC, USA). One-way ANOVA was used for evaluating the effects of reaction time on oil and protein extraction yields at the central point condition. Two-way ANOVA was used for evaluating the effects of different extraction conditions at different reaction times during the experimental validation of best extraction conditions. Multiple comparisons of least-square means were made by Tukey's adjustment with level of significance set at *p* < 0.05. Multivariate statistical analyses were performed using the Protimiza Experimental Design® Software (Protimiza Experimental Design, Campinas, Brazil) and the significance of the effects was determined by the t-test and *p*-value, which was set at *p* < 0.05.

## 3. Results and Discussion

### 3.1. Enzyme Selection for the Enzyme-assisted Extraction Process of Almond Flour

Enzymatic hydrolysis has shown to increase the extraction of oil and protein from many oil-bearing materials, the magnitude of which depending on the type and source of the enzyme used and on the treatments to which the material has been previously subjected to (e.g., mechanical disruption and heat treatments) [12,18,26]. The performance of NP2M and NPL in regards to total oil and protein extractability and distribution of the extracted compounds in each phase generated by the process is shown in Figure 2A,B.

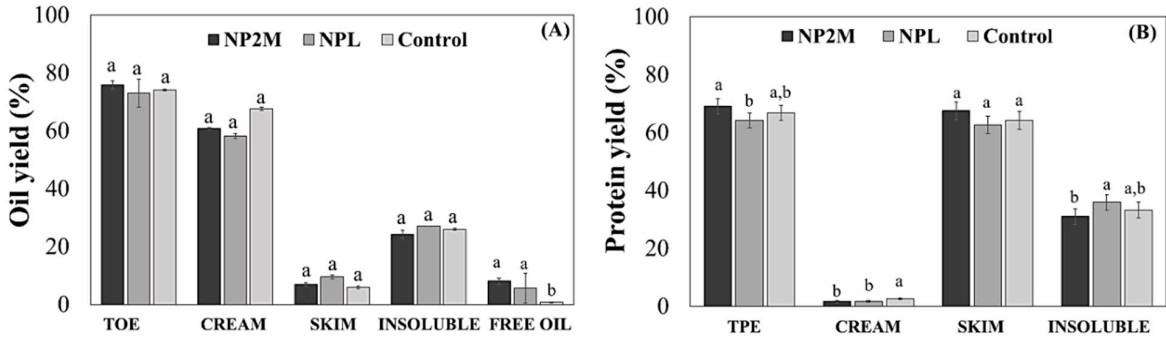

**Figure 2.** Extraction performance of Neutral Protease 2 million (NP2M) and Neutral Protease L (NPL) on oil (**A**) and protein (**B**) extraction yields from almond flour. Data were analyzed by one-way ANOVA followed by Tukey's post-hoc test. Different letters indicate significant difference within each fraction (*p* < 0.05). TPE: Total protein extraction, TOE: Total oil extraction.

The use of proteases to assist the extraction did not lead to significant increment in TOE and TPE yields (Figure 2A,B), compared to the control (without enzyme). Similar oil extraction yields were observed when using both proteases to assist the extraction compared with the control (73–76 vs.

74%) (Figure 2A). However, a significant increase ($p < 0.05$) in free oil yield was observed when using both proteases compared with the control. Free oil yield increased from $0.7 \pm 0.3\%$ (control) to $8 \pm 1\%$ and $6 \pm 4\%$ when NP2M and NPL were used, respectively (Figure 2A). Higher free oil yield when using proteases during the extraction is likely the result of the action of the proteases on the cream protein, which can be further hydrolyzed thus improving the release of the entrapped oil in the cream. The higher amount of free oil obtained when using both proteases is evidenced by the slightly lower oil yield in the cream (58–61%) compared with the control (68%), although this difference was not statistically significant at $p < 0.05$. Reduced oil yield in the skim (6–9%) was observed for all treatments, not being statistically different at $p < 0.05$ (Figure 2A).

As observed for TOE, similar protein extraction yields were obtained by the AEP and EAEP. Protein extraction yields of $64 \pm 3$ and $69 \pm 3\%$ were achieved when NPL and NP2M were used to assist the extraction, respectively (Figure 2B). It is worth mentioning that the use of proteases during the extraction led to a significantly lower protein yield in the cream fraction compared with the control (2 vs. 3%), which might affect the resistance of the EAEP cream towards subsequent de-emulsification (i.e., the release of the entrapped oil from the emulsion). Because the use of NP2M resulted in higher free oil yield and slightly higher protein yield in the skim as evidenced by the reduced amount of protein in the cream, this enzyme was selected for the subsequent study where a broader range of reaction parameters was investigated.

### 3.2. Effects of Extraction Parameters on the Enzyme-assisted Aqueous Extraction Process (EAEP) of Almond Flour

A preliminary evaluation of the extraction kinetics at the central point (pH 8.0, 50 °C, 1:10 SLR, and 0.75% of enzyme) was performed with the objective of identifying the appropriate reaction time range to be evaluated in the fractional factorial design. Extractions were performed for 1, 2, 3, 4, and 5 h. The use of pH 8.0 in the central point, instead of pH 9.0 as used during the enzyme-screening step, had the objective of avoiding the use of excessive alkaline pH (above 10) in the + 1 level in the fractional factorial design. As observed in Figure 3A,B, increasing extraction time from 1 to 5 h did not significantly improve oil and protein extraction yields at $p < 0.05$. Based on these results, the effects of the variables ranging from levels −1 and +1 on extraction yields of oil and protein were evaluated at 40 min and 180 min (3 h). Reaction time of 40 min was selected to evaluate if similar extraction yields could be achieved at shorter reaction times (<1 h), while 3 h was selected as a conservative strategy to further confirm the trend observed.

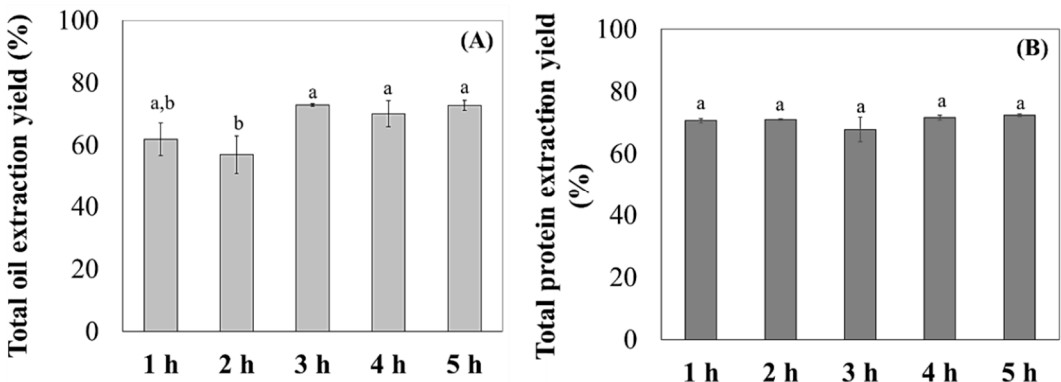

**Figure 3.** Oil (**A**) and protein (**B**) extraction yields at the central point (pH 8.0, 50 °C, 1:10 SLR, and 0.75% (*w/w*) of NP2M). Data were analyzed by one-way ANOVA followed by Tukey's post-hoc test. Different letters indicate a significant difference between samples at $p < 0.05$.

Oil extraction yields and its distribution among the fractions generated for each experimental condition evaluated in the fractional factorial are described in Figure 4A–E. Higher oil extraction yields were observed for experiments performed at the central point conditions (#9–11) (pH 8.0, 50 °C, 1:10

SLR, 0.75% enzyme). No significant increment in oil extractability was observed when reaction time increased from 40 min to 3 h. Total oil extraction yields of 71 ± 4% and 73 ± 1% were achieved at 40 and 180 min, respectively (Figure 4A). In addition to achieving high extraction yields, the form in which the extracted oil is present (i.e., as free oil or entrapped in the cream fraction) is of great relevance when considering subsequent recovery and utilization of the extracted oil. At 40 min, higher free oil yields (8–16%) were observed at experiments #2 (pH 9.5, 45 °C, 1:12 SLR, 1.0% enzyme), #3 (pH 6.5, 55 °C, 1:12 SLR, 1.0% enzyme), and #4 (pH 9.5, 55 °C, 1:12 SLR, 0.5% enzyme) (Figure 4E). Overall, free oil yield decreased when extraction time increased from 40 to 180 min, corresponding to an increase in the amount of oil in the cream fraction (Figure 4B) and reduction in the amount of oil yield in the skim (Figure 4C). These results indicate that emulsion formation was favored with increasing reaction time, thus leading to less free oil and more oil entrapped in the cream fraction.

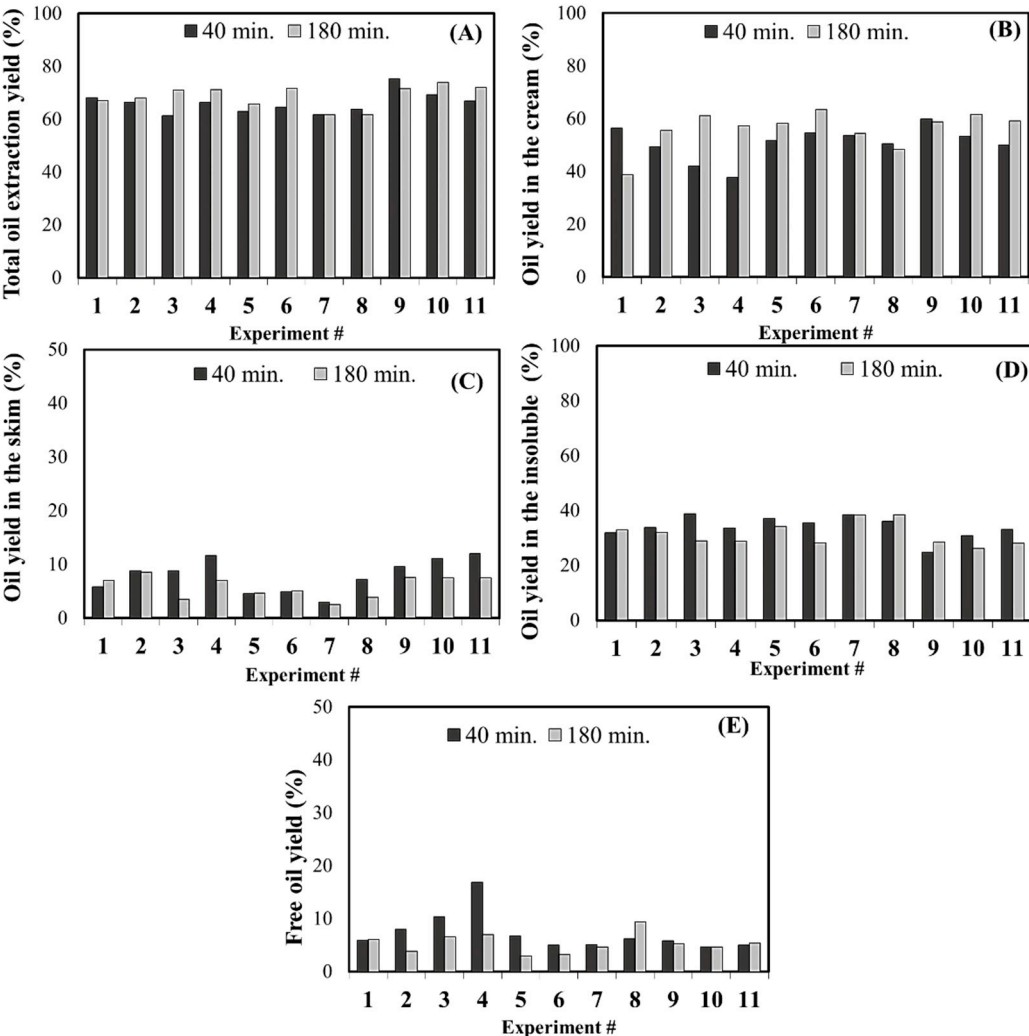

**Figure 4.** Oil extraction yields and distribution for each experimental condition evaluated in the fractional factorial design: TOE yield (%) (**A**), oil yield in the cream (%) (**B**), oil yield in the skim (%) (**C**), oil yield in the insoluble (%) (**D**), and free oil (%) (**E**). Experimental conditions: #1 (pH 6.5, 45 °C, 1:12 SLR, 0.50% enzyme (*w/w\**)); #2 (pH 9.5, 45 °C, 1:12 SLR, 1.0% enzyme (*w/w*)); #3 (pH 6.5, 55 °C, 1:12 SLR, 1.0% enzyme (*w/w*)); #4 (pH 9.5, 55 °C, 1:12 SLR, 0.50% enzyme (*w/w*)); #5 (pH 6.5, 45 °C, 1:8 SLR, 1.00% enzyme (*w/w*)); #6 (pH 9.5, 45 °C, 1:8 SLR, 0.50% enzyme (*w/w*)); #7 (pH 6.5, 55 °C, 1:8 SLR, 0.50% enzyme (*w/w*)); #8 (pH 9.5, 55 °C, 1:8 SLR, 1.00% enzyme (*w/w*)); #9, 10 and 11 (pH 8.0, 50 °C, 1:10 SLR, 0.75% enzyme (*w/w*)); * *w/w*, weight of enzyme/weight of almond flour.

The estimated effects of the variables evaluated (pH, temperature, SLR, and enzyme) on oil extraction yields as well as on the distribution of the extracted compounds among the fractions are shown in Table 2.

**Table 2.** Estimated effects of pH, temperature, SLR, and amount of enzyme on oil extraction yields at 40 and 180 min of extraction.

| | Oil Extraction | | | | | | | | | |
|---|---|---|---|---|---|---|---|---|---|---|
| | Extraction Time = 40 min | | | | | | | | | |
| Factors | Effect | *p*-Value | Effect | *p*-Value | Effect | *p*-Value | Effect | *p*-Value | Effect | *p*-Value |
| | TOE | | OYC | | OYS | | OYI | | FOY | |
| Mean | 64.35 | 0.0000 | 49.51 | 0.0000 | 6.84 | 0.0000 | 35.64 | 0.0000 | 8.00 | 0.0006 |
| Curvature | 12.18 | 0.0340 | 9.77 | 0.2295 | 8.11 | 0.0053 | −12.17 | 0.0341 | −5.69 | 0.2066 |
| pH ($x_1$) | 1.76 | 0.4599 | −2.83 | 0.4816 | 2.60 | 0.0341 | −1.76 | 0.4600 | 1.99 | 0.3757 |
| Temperature (°C) ($x_2$) | −2.17 | 0.3686 | −7.05 | 0.1172 | 1.64 | 0.1277 | 2.17 | 0.3681 | 3.24 | 0.1745 |
| SLR* ($x_3$) | −2.25 | 0.3520 | 6.19 | 0.1575 | −3.89 | 0.0076 | 2.25 | 0.3522 | −4.56 | 0.0766 |
| Enzyme (%) (*w/w*) ($x_4$) | −1.53 | 0.5166 | −2.12 | 0.5937 | 1.02 | 0.3100 | 1.53 | 0.5165 | −0.43 | 0.8412 |
| | Extraction Time = 180 min | | | | | | | | | |
| Factors | TOE | | OYC | | OYS | | OYI | | FOY | |
| | Effect | *p*-Value | Effect | *p*-Value | Effect | *p*-Value | Effect | *p*-Value | Effect | *p*-Value |
| Mean | 67.27 | 0.0000 | 54.64 | 0.0000 | 7.18 | 0.0057 | 32.73 | 0.0000 | 5.46 | 0.0003 |
| Curvature | 10.39 | 0.0934 | 10.43 | 0.4224 | 0.66 | 0.9161 | −10.39 | 0.0934 | −0.71 | 0.7748 |
| pH ($x_1$) | 1.69 | 0.5472 | 3.01 | 0.6496 | −2.14 | 0.5204 | −1.69 | 0.5473 | 0.82 | 0.5302 |
| Temperature (°C) ($x_2$) | −1.74 | 0.5359 | 1.28 | 0.8448 | −5.91 | 0.1151 | 1.74 | 0.5360 | 2.88 | 0.0646 |
| SLR* ($x_3$) | −4.12 | 0.1769 | 2.97 | 0.6544 | −6.31 | 0.0976 | 4.12 | 0.1769 | −0.78 | 0.5521 |
| Enzyme (%) (*w/w*) ($x_4$) | −1.37 | 0.6244 | 2.31 | 0.7261 | −4.14 | 0.2391 | 1.37 | 0.6244 | 0.46 | 0.7202 |

TOE: Total oil extraction; OYC: oil yield in the cream; OYS: oil yield in the skim; OYI: oil yield in the insoluble; FOY: Free oil yield. * SLR: Solid-to-liquid-ratio.

The estimated effects of the extraction parameters evaluated on the extraction and separation of oil from almond flour are shown in Table 2. Overall, total oil extraction was not significantly affected by the reaction parameters in the range evaluated, regardless of the extraction time evaluated. According to the effects described in Table 2, reduced oil yield in the skim could be achieved by the use of higher SLR (1:8) (effect value −3.89, $p < 0.05$) and lower pH values (effect value of 2.6, $p < 0.05$), at 40 min of extraction. The curvature analysis revealed that the curvature for TOE and OYS at 40 min of extraction time was significant and positive, indicating that higher yields were observed around the central point conditions. For all responses evaluated, the experiments performed at the central point conditions (experiments #9–11) presented a low relative standard deviation (1 to 8%), indicating good reproducibility of the experimental data.

Protein extraction yields and its distribution among the fractions generated by the process (skim, cream, insoluble) for all experimental conditions evaluated in the fractional factorial are described in Figure 5A–D.

At 40 min of extraction, higher protein extraction yields were observed for experiments #7 (79%) (pH 6.5, 55 °C, 1:8 SLR, 0.5% enzyme), # 3 (75%) (pH 6.5, 55 °C, 1:12 SLR, 1.0% enzyme), and experiments performed at the central point (71 ± 1%) (pH 8.0, 50 °C, 1:10 SLR, 0.75% enzyme) (Figure 5A). At 180 min of extraction, higher protein extraction yields were observed at the central point conditions (experiments # 9–11) (76 ± 2%) (pH 8.0, 50 °C, 1:10 SLR, 0.75% enzyme), followed by experiments # 2 (75%) (pH 9.5, 45 °C, 1:12 SLR, 1.0% enzyme) and # 3 (73%) (pH 6.5, 55 °C, 1:12 SLR, 1.0% enzyme) (Figure 5A). As expected, a higher protein yield in the skims generated by these treatments was observed (Figure 5C). Protein yields from 69–79% and 71–75% were observed in the skims obtained from the experiments above described at 40 and 180 min, respectively. Low protein yield in the cream was observed for all experimental conditions, with values ranging from 0.4 to 4% (Figure 5B). The estimated effects of the variables evaluated (pH, temperature, SLR, and enzyme) on protein extractability and distribution of the extracted protein among the fractions are shown in Table 3.

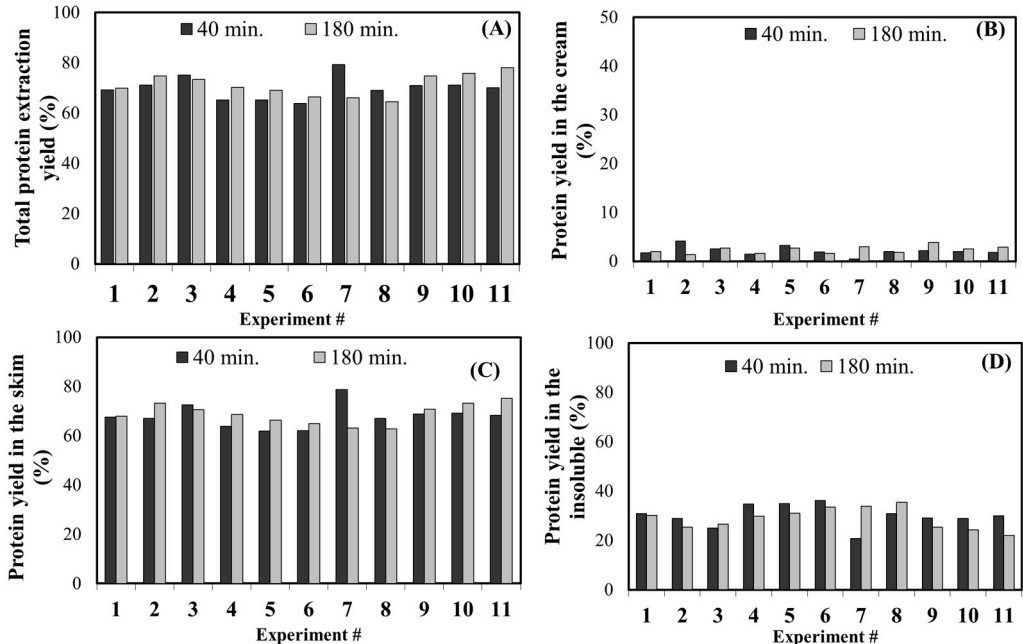

**Figure 5.** Protein extraction yields and distribution for each experimental condition evaluated in the fractional factorial design: total protein extraction yields (%) (**A**), protein yield in the cream (%) (**B**), protein yield in the skim (%) (**C**), and protein yield in the insolubles (%) (**D**). Experimental conditions: #1 (pH 6.5, 45 °C, 1:12 SLR, 0.50% enzyme (*w/w*\*)); #2 (pH 9.5, 45 °C, 1:12 SLR, 1.0% enzyme (*w/w*)); #3 (pH 6.5, 55 °C, 1:12 SLR, 1.0% enzyme (*w/w*)); #4 (pH 9.5, 55 °C, 1:12 SLR, 0.50% enzyme (*w/w*)); #5 (pH 6.5, 45 °C, 1:8 SLR, 1.00% enzyme (*w/w*)); #6 (pH 9.5, 45 °C, 1:8 SLR, 0.50% enzyme (*w/w*)); #7 (pH 6.5, 55 °C, 1:8 SLR, 0.50% enzyme (*w/w*)); #8 (pH 9.5, 55 °C, 1:8 SLR, 1.0% enzyme (*w/w*)); #9, 10 and 11 (pH 8.0, 50 °C, 1:10 SLR, 0.75% enzyme (*w/w*)); \* *w/w*, weight of enzyme/weight of almond flour. Experiments were randomly performed.

**Table 3.** Estimated effects of pH, temperature, SLR and amount of enzyme on protein extraction yields at 40 and 180 min.

| | | | | | | | | |
|---|---|---|---|---|---|---|---|---|
| **Protein Extraction** | | | | | | | | |
| **Extraction Time = 40 min** | | | | | | | | |
| **Factors** | **TPE** | | **PYC** | | **PYS** | | **PYI** | |
| | **Effect** | **p-Value** | **Effect** | **p-Value** | **Effect** | **p-Value** | **Effect** | **p-Value** |
| Mean | 69.75 | 0.0000 | 2.16 | 0.0000 | 67.59 | 0.0000 | 30.25 | 0.0000 |
| Curvature | 1.86 | 0.7727 | −0.36 | 0.3783 | 2.22 | 0.7314 | −1.86 | 0.7729 |
| pH ($x_1$) | −4.83 | 0.1892 | 0.38 | 0.1121 | −5.21 | 0.1633 | 4.83 | 0.1890 |
| Temperature (°C) ($x_2$) | 4.81 | 0.1913 | −1.16 | 0.0020 | 5.97 | 0.1206 | −4.81 | 0.1913 |
| SLR\* ($x_3$) | −0.84 | 0.8014 | −0.58 | 0.0332 | −0.27 | 0.9369 | 0.85 | 0.8010 |
| Enzyme (%) (*w/w*) ($x_4$) | 0.71 | 0.8312 | 1.59 | 0.0005 | −0.88 | 0.7948 | −0.71 | 0.8312 |
| **Extraction Time = 180 min** | | | | | | | | |
| **Factors** | **TPE** | | **PYC** | | **PYS** | | **PYI** | |
| | **Effect** | **p-Value** | **Effect** | **p-Value** | **Effect** | **p-Value** | **Effect** | **p-Value** |
| Mean | 69.27 | 0.0000 | 2.09 | 0.0001 | 67.19 | 0.0000 | 30.73 | 0.0000 |
| Curvature | 13.78 | 0.0031 | 2.02 | 0.0248 | 11.75 | 0.0072 | −13.78 | 0.0031 |
| pH ($x_1$) | −0.58 | 0.6838 | −1.02 | 0.0283 | 0.44 | 0.7679 | 0.58 | 0.6838 |
| Temperature (°C) ($x_2$) | −1.46 | 0.3291 | 0.35 | 0.3383 | −1.81 | 0.2531 | 1.46 | 0.3291 |
| SLR\* ($x_3$) | −5.50 | 0.0096 | 0.36 | 0.3260 | −5.86 | 0.0087 | 5.50 | 0.0096 |
| Enzyme (%) (*w/w*) ($x_4$) | 2.23 | 0.1595 | 0.09 | 0.7923 | 2.13 | 0.1892 | −2.23 | 0.1595 |

TPE: Total protein extraction. PYC: protein yield in the cream. PYS: protein yield in the skim. PIY: protein yield in the insoluble, \* SLR: Solids-to-liquid-ratio.

In general, higher TPE yields (69–75% at 40 min and 70–78% at 180 min) were observed at lower SLR (1:12 and 1:10), with minimum increment in protein extractability observed when extraction time increased from 40 to 180 min. Although SLR had a negative effect for total protein extraction at 40 (−0.84) and 180 min (−5.50), indicating reduced protein extraction at higher SLR, this effect was statistically significant ($p < 0.05$) only at 180 min (Table 3). As the result of increased protein extractability at low SLR at 180 min, SLR had a negative and significant effect on protein yield in the skim, indicating that the use of a low SLR would favor total protein extraction thus leading to a skim fraction with higher protein content. Increased protein extractability at lower SLR is commonly attributed to increased gradient concentration between solutes and the aqueous medium, which reduces the viscosity of the extraction medium thus favoring protein diffusion into the extraction medium [27]. However, increased SLR led to a reduction in the protein yield in the cream as demonstrated by the negative and significant effect (−0.58) ($p < 0.05$) at 40 min (Table 3). Overall, our results are in agreement with the ones presented by Rosenthal et al. [18], De Moura et al. [11] and Souza et al. [16] in which higher protein extraction yields have been achieved by the use of lower SLR. The comparison of our results with the ones in the literature is challenging due to the differences in the starting material used. However, a similar trend has been observed by Souza et al. [16], which reported significant increment in protein extractability from the almond press cake (41 to 47% for the AEP and 60 to 75% for the EAEP) when SLR was reduced from 1:7.18 to 1:12.82. Similarly, Esteban et al. [28] reported higher protein extractability when using a solvent/almond meal ratio of 2000:1 (vol/wt) for a single extraction. Despite higher protein extractability at low SLR, it is worth mentioning the importance of finding a compromise between the amount of solvent used and the target extractability. The use of a higher amount of solvent results in a higher amount of effluent for subsequent fractionation.

When evaluating the effects of the use of enzyme on protein extractability, it can be observed that, except by the higher protein yield in the cream at 40 min ($p < 0.05$), increasing the amount of enzyme from 0.5 to 1.0% did not result in a significant increase in protein extractability nor shifted the protein distribution among the fractions, regardless the extraction time used (Table 3). Within the range evaluated, our results indicate that the use of 0.5% enzyme is sufficient to achieve high protein extraction yields.

Increasing the slurry pH from 6.5 to 9.0 did not significantly alter protein extractability, regardless of the extraction time evaluated. Taking into account that the enzyme used in this study is stable at pH values ranging from 5.5 to 9.0 and that almond proteins have a higher solubility at higher pH [29,30], increased protein extractability at higher pH would be expected. In addition to its effect on overall extractability, pH can also impact the extracted protein functionality [31], as described in Section 3.4. Therefore, extraction yields should not be the only response evaluated when selecting the best extraction conditions. At 180 min, the amount of protein in the cream fraction was significantly affected by pH (−1.02), the negative effect indicating that when pH increases from 6.5 to 9.0, the protein yield in the cream decreases. Considering the well-known ability of proteins to interact with oil and to form stable emulsions [32], it becomes necessary to understand the effects of the cream composition regarding its resistance to subsequent de-emulsification, which is beyond the scope of this work.

Increasing the extraction temperature from 45 to 55 °C did not significantly increase TPE, regardless of the extraction time evaluated. While a positive trend was observed when using higher temperatures at 40 min, a negative trend was observed at 180 min, which could be due to possible enzyme or protein denaturation when exposed at 55 °C for longer times. However, these effects were not statistically significant at $p < 0.05$. Our results are in agreement with the ones presented by Esteban et al. [28] which reported that a temperature of 50 °C increased the solubility of almond protein while the use of higher temperatures (50–75 °C) could promote partial coagulation of the protein which would, in turn, reduce its solubility.

Overall, the effects observed indicate that lower values of temperature, SLR (higher amount of water) and amount of enzyme could result in the production of a cream fraction with a lower protein content at 40 min. Since proteins can act as strong emulsifiers [33], a lower protein concentration in the

cream might be desirable when considering the subsequent de-emulsification step to which the cream needs to be subjected to release the entrapped oil. The same conditions leading to reduced protein yield in the cream could be exploited to achieve high protein extraction yields at 40 min at any pH value in the range evaluated.

Overall, the effects described in Tables 2 and 3 indicate that all processing variables (pH, temperature, SLR, and enzyme) could be used at their lowest level (−1.0) to achieve high oil and protein extraction yields at 40 min. However, the use of a higher SLR (1:10 or 1:8) could enable the production of a cream fraction with lower protein content and a skim fraction with lower oil content. In order to further confirm the observed effects of the extraction parameters evaluated, extraction conditions were replicated as described in the section below.

*3.3. Experimental Validation of the Best Extraction Conditions*

Because increasing reaction time from 40 to 180 min did not result in higher extractability, the validation of the best conditions identified in the fractional factorial design was performed at 20, 40 and 60 min to maximize process productivity. Based on the estimated effects (Tables 2 and 3) and preliminary conditions used during the enzyme screening, two experimental conditions were selected for the validation experiments, including a control experiment for each treatment (AEP, without enzyme). Experimental conditions and treatments are described as the following:

(i)    Condition 1: pH 6.5, 1:8 SLR, 45 °C and 0.5% of enzyme for the EAEP1, and without enzyme for the AEP1.

(ii)   Condition 2: pH 9.0, 1:10 SLR, 50 °C and 0.5% of enzyme for the EAEP2, and without enzyme for the AEP2.

Extraction yields and distribution of the extracted compounds for the $EAEP_1$ and $EAEP_2$, with their respective controls, are shown in Tables 4 and 5. Total oil extraction did not significantly increase when extraction time increased from 20 to 60 min for $EAEP_1$ (pH 6.5, 0.5% NP2M, 1:8 SLR, and 45 °C) (56 ± 2% vs. 61 ± 2 %) (Table 4). The use of a more alkaline pH (9.0), an intermediate SLR (1:10), and a slightly higher temperature (50 °C) ($EAEP_2$) led to higher oil extraction yields compared to the ones obtained in $EAEP_1$ (75% for $EAEP_2$ vs. 61% $EAEP_1$). However, similar oil extraction yields were observed within each condition, with or without enzyme use, indicating that the use of enzyme did not play a key role in improving oil extraction yields compared with the other parameters evaluated such as pH. Overall, increasing reaction time from 20 to 60 min significantly increased total oil extraction for the $EAEP_2$ (62 vs. 75%).

The highest oil yield in the skim (7.8 ± 2.0%) was observed for $EAEP_1$ (pH 6.5, 0.5% NP2 M, 1:8 SLR, and 45 °C) at 60 min, being statistically higher than the 4.0% observed for the control experiment ($AEP_1$). Slightly lower oil yields in the skim (4.7 and 4.3%) were achieved for $EAEP_2$ (pH 9.0, 0.5% N2 MP, 1:10 SLR, and 50 °C) and $AEP_2$, respectively. It is worth mentioning that the separation of the fractions (skim, cream, and free oil) at lower pH (6.5) is challenging compared with the use of pH 9.0, which might be related to the higher oil content of the skim fractions produced by the $EAEP_1$. Overall, cream fractions with higher oil yields were obtained from the $EAEP_2$ (63–66%) compared with the $EAEP_1$ (51–47%). Similar free oil yields (3–6%) were obtained, regardless of the treatment used.

Protein extraction yields increased from 55 ± 5 to 68 ± 5 % when reaction time increased from 20 to 60 min for the $EAEP_1$ (Table 5), being significantly higher than the 55 ± 4% obtained for the control experiment at 60 min ($AEP_1$). Overall, high protein extraction yields were achieved at shorter reaction times for the $EAEP_2$ compared with the $EAEP_1$ (70 vs. 55% at 20 min and 72 vs. 63% at 40 min). For the $EAEP_2$ and its control ($AEP_2$), increasing extraction time from 20 to 60 min did not significantly increase protein extraction. Our results evidence the key role of extraction pH on overall protein extractability. At lower pH (6.5), where protein solubilization is not as favored as compared with a more alkaline pH (9.0) [31], the use of enzyme has a positive effect on improving extraction yields compared with the control. However, at alkaline pH (9.0), where almond protein solubilization

is favored, the use of enzyme does not play a key role in overall protein extractability. Based on the validation experiments, high protein and oil extraction yields (70%) could be achieved in short reaction time when using pH 9.0, 1:10 SLR, and 50 °C. While protein extraction yields of approximately 70% could be achieved at 20 min, similar oil extraction yields would require 40–60 min of extraction.

**Table 4.** Oil extraction yields for the validation experiments of the enzyme-assisted aqueous extraction process $EAEP_1$ and $EAEP_2$.

| Total Oil Extraction Yield (%) | | | |
|---|---|---|---|
| **Samples** | **20 min** | **40 min** | **60 min** |
| $EAEP_1$ | 55.7 ± 1.6 [e.f] | 54.5 ± 3.4 [e.f] | 61.1 ± 1.7 [c.d.e] |
| $AEP_1$ | 48.1 ± 2.4 [f] | 56.1 ± 2.8 [e.f] | 58.3 ± 4.7 [d.e] |
| $EAEP_2$ | 62.0 ± 0.7 [b.c.d.e] | 68.6 ± 3.3 [a.b.c] | 74.5 ± 2.2 [a] |
| $AEP_2$ | 64.9 ± 2.6 [b.c.d] | 66.9 ± 5.1 [a.b.c] | 69.8 ± 0.9 [a.b] |
| **Oil Yield in the Skim (%)** | | | |
| **Samples** | **20 min** | **40 min** | **60 min** |
| $EAEP_1$ | 2.5 ± 0.5 [c] | 3.8 ± 1.6 [b.c] | 7.8 ± 2.0 [a] |
| $AEP_1$ | 3.7 ± 1.0 [b.c] | 4.6 ± 1.9 [a.b.c] | 4.0 ± 1.0 [b.c] |
| $EAEP_2$ | 5.0 ± 1.0 [a.b.c] | 5.4 ± 1.3 [a.b.c] | 4.7 ± 1.8 [a.b.c] |
| $AEP_2$ | 6.5 ± 0.6 [a.b] | 4.8 ± 0.9 [a.b.c] | 4.30 ± 0.9 [a.b.c] |
| **Oil Yield in the Cream (%)** | | | |
| **Samples** | **20 min** | **40 min** | **60 min** |
| $EAEP_1$ | 44.6 ± 3.2 [e.f] | 45.0 ± 3.8 [e.f] | 47.0 ± 3.7 [d.e.f] |
| $AEP_1$ | 41.0 ± 1.1 [f] | 48.2 ± 4.0 [c.d.e.f] | 51.0 ± 3.7 [b.c.d.e.f] |
| $EAEP_2$ | 52.3 ± 1.5 [b.c.d.e] | 57.3 ± 4.8 [a.b.c] | 65.8 ± 4.4 [a] |
| $AEP_2$ | 55.9 ± 3.3 [a.b.c.d] | 59.0 ± 6.1 [a.b] | 63.37 ± 1.6 [a] |
| **Oil Yield in the Insoluble (%)** | | | |
| **Samples** | **20 min** | **40 min** | **60 min** |
| $EAEP_1$ | 44.3 ± 1.6 [a.b] | 45.6 ± 3.4 [a.b] | 38.9 ± 1.7 [b.c.d] |
| $AEP_1$ | 51.9 ± 2.4 [a] | 43.9 ± 2.8 [a.b] | 41.7 ± 4.7 [b.c] |
| $EAEP_2$ | 38.0 ± 0.7 [b.c.d.e] | 31.4 ± 3.3 [d.e.f] | 25.5 ± 2.2 [f] |
| $AEP_2$ | 35.1 ± 2.6 [c.d.e] | 33.1 ± 5 [d.e.f] | 29.8 ± 1.3 [e.f] |
| **Free Oil Yield (%)** | | | |
| **Samples** | **20 min** | **40 min** | **60 min** |
| $EAEP_1$ | 8.7 ± 1.5 [a] | 5.5 ± 1.7 [a.b.c.d] | 6.4 ± 1.0 [a.b] |
| $AEP_1$ | 3.3 ± 1.7 [b.c.d] | 3.3 ± 0.5 [b.c.d] | 3.3 ± 0.2 [b.c.d] |
| $EAEP_2$ | 4.6 ± 1.5 [b.c.d] | 5.9 ± 2.3 [a.b.c] | 6.4 ± 0.5 [a.b] |
| $AEP_2$ | 2.5 ± 0.4 [d] | 3.2 ± 0.7 [b.c.d] | 2.6 ± 0.5 [c.d] |

Means with different lowercase letters indicate statistical difference at $p < 0.05$. Statistical significance differences were denoted by different letters, with the letter "a" being assigned to the highest value. $EAEP_1$: pH 6.5, 0.5% of enzyme, 1:8 SLR, and 45 °C; $EAEP_1$ control ($AEP_1$): pH 6.5, 1:8 SLR, and 45 °C; $EAEP_2$: pH 9.0, 0.5% of enzyme, 1:10 SLR, and 50 °C; $EAEP_2$ control ($AEP_2$): pH 9.0, 1:10 SLR, and 50 °C.

The comparison of our data with the ones in the literature is challenging due to differences in the starting materials used and lack of studies focusing on the simultaneous extraction of oil and protein from almond flour. Oil extraction yields obtained in our study (75% for $EAEP_2$ and 70% for $AEP_2$) are similar or higher to the ones reported by Sharma and Gupta [14], where oil extraction yields ranging from 37–40% were reported for the AEP (pH 4–9, 18 h reaction time) and 75–77% for the EAEP process (40 °C, 4–18 h, and use of 416 U of the enzyme Protizyme TM). Balvardi et al. [13] reported a 77.8% oil extraction recovery from almond flour when extractions were performed at pH 5.76, 50 °C, reaction time of 4 h, and a mixture of proteases and cellulases was used at 1% level. However, in their study oil recovery yields were calculated in relation to the ones obtained using hexane extraction, while herein

extraction yields were calculated in relation to the initial amount of oil present in the almond flour that can be extracted, making difficult to make a direct comparison of the results from both studies. No protein extraction yields were reported for both studies. The simultaneous extraction of oil and protein from the almond press cake has been recently reported by Souza et al. [16]. In their study, oil extraction yields of 42 and 50% were achieved for the AEP and EAEP, respectively. In regards to protein extraction, 70 and 75% of the almond cake protein was achieved for the AEP and EAEP, respectively. However, it is worth mentioning the use of almond press cake in their study and almond flour in ours.

**Table 5.** Protein extraction yields for the validation experiments of the $EAEP_1$ and $EAEP_2$.

| Total Protein Extraction Yield (%) | | | |
|---|---|---|---|
| **Samples** | **20 min** | **40 min** | **60 min** |
| $EAEP_1$ | 55.1 ± 4.7 [c] | 63.3 ± 2.8 [b.c] | 68.1 ± 5.2 [a.b] |
| $AEP_1$ | 55.7 ±2.6 [c] | 56.0 ± 0.5 [c] | 55.3 ± 3.9 [c] |
| $EAEP_2$ | 69.3 ± 1.2 [a.b] | 71.9± 2.4 [a] | 72.3 ± 0.9 [a] |
| $AEP_2$ | 65.0 ± 0.8 [a.b] | 66.5 ± 1.6 [a.b] | 64.8± 2.2 [a.b] |
| **Protein Yield in the Skim (%)** | | | |
| **Samples** | **20 min** | **40 min** | **60 min** |
| $EAEP_1$ | 50.8 ± 5.3 [d] | 60.6 ± 2.1 [b.c] | 63.5 ± 4.0 [a.b] |
| $AEP_1$ | 53.0 ± 2.9 [c.d] | 53.3 ± 0.8 [c.d] | 51.7 ± 4.4 [d] |
| $EAEP_2$ | 67.7 ± 1.1 [a.b] | 70.4 ± 2.2 [a] | 70.94 ± 0.9 [a] |
| $AEP_2$ | 63.2 ± 0.6 [a.b] | 64.6 ± 1.5 [a.b] | 65.7 ± 2.2 [a.b.c] |
| **Protein Yield in the Cream (%)** | | | |
| $EAEP_1$ | 4.3 ± 1.4 [a] | 2.7 ± 0.6 [a] | 4.5 ± 2.1 [a] |
| $AEP_1$ | 2.7 ± 0.4 [a] | 2.7 ± 0.6 [a] | 3.6 ± 0.5 [a] |
| $EAEP_2$ | 1.6 ± 0.1 [a] | 1.5 ± 0.1 [a] | 1.6 ± 0.1 [a] |
| $AEP_2$ | 1.8 ± 0.2 [a] | 1.9 ± 0.2 [a] | 2.0 ± 0.3 [a] |

Means with different lowercase letters indicate statistical difference at $p < 0.05$. Statistical significance differences were denoted by different letters, with the letter "a" being assigned to the highest value $EAEP_1$: pH 6.5, 0.5% of enzyme, 1:8 SLR, and 45 °C; $EAEP_1$ control ($AEP_1$): pH 6.5, 1:8 SLR, and 45 °C; $EAEP_2$: pH 9.0, 0.5% of enzyme, 1:10 SLR, and 50 °C; $EAEP_2$ control ($AEP_2$): pH 9.0, 1:10 SLR, and 50 °C.

Overall, our results demonstrate that experimental validation of the best conditions identified in the fractional factorial design enabled the identification of processing parameters leading to high extraction yields of oil and protein at very short extraction time compared with existing data in the literature (40–60 min vs. 4–18 h). However, because similar extraction yields could be achieved at different reaction conditions (acidic and alkaline pH, with and without enzyme), although at the expense of longer reaction times, the impact of the best extraction conditions on the extracted protein solubility was evaluated.

### 3.4. Molecular Weight Polypeptide Profile, Degree of Hydrolysis (DH) and Solubility of AEP and EAEP Skims

The use of proteases during the extraction process leads to the breakdown of peptide bonds thus resulting in increased concentration of primary amines which corresponds to an increase in the degree of hydrolysis (DH) [34]. The functionality of the protein hydrolysate is tied to the nature and the composition of peptides generated during hydrolysis [35]. Therefore, understanding how extraction conditions impact the DH and protein functionality becomes necessary to better select processing conditions that will lead to the production of proteins with desired technological functions.

Figure 6 shows the impact of the extraction conditions evaluated during the experimental validation of the AEP and EAEP on the DH and protein profile of AEP and EAEP skims. As expected, no changes in the DH of AEP skim proteins were observed (2%) when reaction time increased from 20 to 60 min for Conditions 1 and 2 (Figure 6). However, extraction time had a significant impact on the DH for the EAEP skim. The DH of the $EAEP_1$ (pH 6.5, 0.5% enzyme, 1:8 SLR, 45 °C) and $EAEP_2$

(pH 9.0, 0.5% enzyme, 1:10 SLR, 50 °C) skim proteins increased from 18 to 25% and 7 to 12% when reaction time increased from 20 to 60 min, respectively. The higher DH of the EAEP$_1$ skim is likely due to the higher enzyme activity at pH 6.5 (95% of its maximum activity), compared with its activity at pH 9.0 (50% of its maximum activity, as described by the manufacturer). Our results are in agreement with the literature where the use of enzyme for a longer time, until a certain extent, resulted in higher DH [11,16,34].

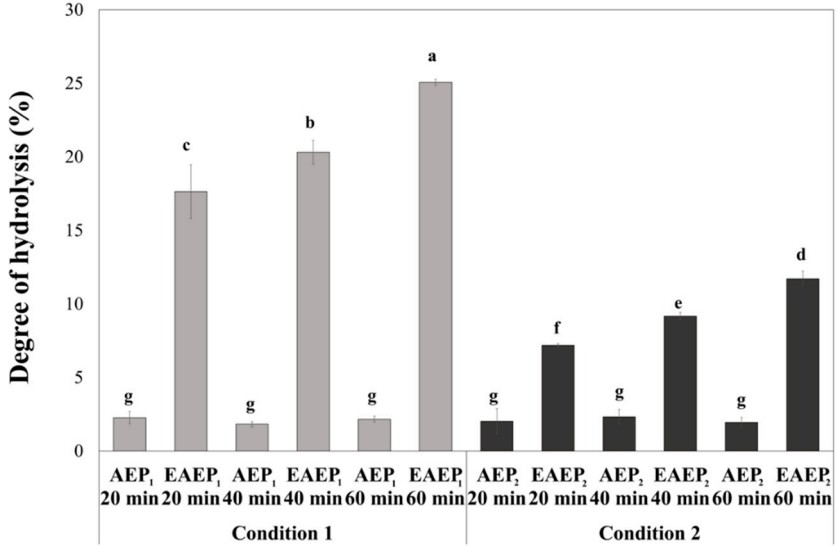

**Figure 6.** Impact of the extraction conditions evaluated during the experimental validation of the AEP and EAEP on the degree of hydrolysis (%) of skim proteins. Data were analyzed by one-way ANOVA followed by Tukey's post-hoc test. Different letters indicate a significant difference between samples at $p < 0.05$. Condition 1: pH 6.5, 0.5% enzyme, 1:8 SLR, and 45 °C for the EAEP$_1$ and pH 6.5, 1:8 SLR, and 45 °C for the AEP$_1$. Condition 2: pH 9.0, 0.5% enzyme, 1:10 SLR, and 50 °C for the EAEP$_2$ and pH 9.0, 1:10 SLR, 50 °C for the AEP$_2$.

The DH values described in Figure 6 are in agreement with the SDS-PAGE peptide profile of AEP and EAEP skim proteins (Figure 7). Similar electrophoretic profile was observed for AEP skim proteins, regardless of the extraction condition used. This trend is in agreement with the constant DH of these samples (2%). AEP skim proteins are mainly composed of two polypeptide fragments (40 kDa (25%, α-subunit) and 24 kDa (30%, β-subunit)) which correspond to the subunits of amandin, the major protein accounting for 65–70% of extractable almond protein [36] (Figure 7). Our results are in agreement with the ones reported by Wolf and Sathe [29] and Derbyshire et al. [37], which demonstrated that amandin (62 to 66 kDa subunits) could be converted into acidic (20 kDa) and basic (40 kDa) polypeptides in the presence of mercaptoethanol. The EAEP skim protein profile evidences increased protein hydrolysis as extraction time increased from 20 to 60 min for both conditions evaluated (from 18 to 25% for EAEP$_1$ and from 7 to 13% for EAEP$_2$). At 60 min of extraction, the EAEP$_1$ showed the highest amount of peptides < 14.4 kDa (80%), indicating that the major polypeptides were broken down into smaller fractions.

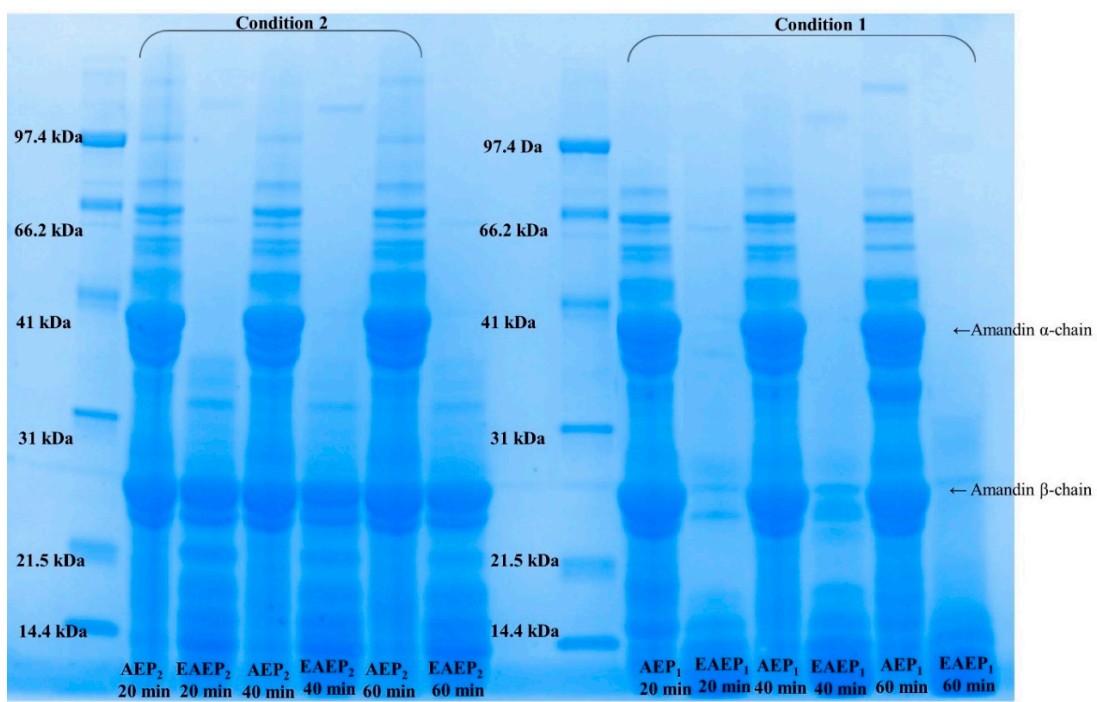

**Figure 7.** SDS-PAGE protein profile of the skims from AEP and EAEP process. Condition 1: pH 6.5, 0.5% enzyme, 1:8 SLR, and 45 °C for the EAEP$_1$ and pH 6.5, 1:8 SLR, and 45 °C for the AEP$_1$. Condition 2: pH 9.0, 0.5% enzyme, 1:10 SLR, and 50 °C for the EAEP$_2$ and pH 9.0, 1:10 SLR, 50 °C for the AEP$_2$.

Since extraction conditions such as pH, enzyme use, and reaction time can significantly affect protein functionality [5,16], we determined how the extraction conditions evaluated during the experimental validation of the AEP and EAEP (Conditions 1 and 2) affected the solubility of the AEP and EAEP skim proteins. With the goal of identifying possible applications for the extracted protein, skim protein solubility was assessed at acidic pH (5.0, isoelectric point for almond protein [37]) and alkaline pH (9.0, extraction pH) (Figure 8A,B).

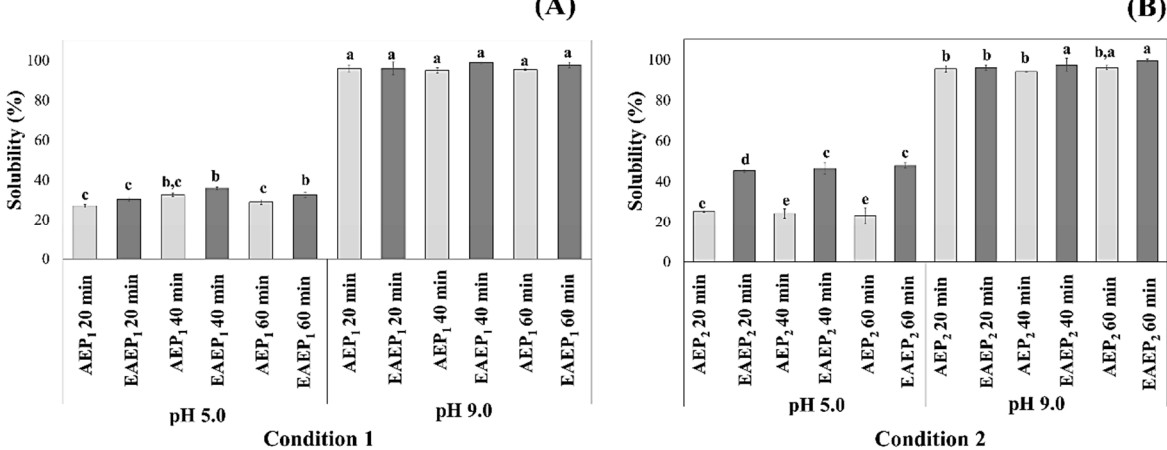

**Figure 8.** Impact of extraction conditions evaluated during the experimental validation of the AEP and EAEP on the solubility of AEP and EAEP skim proteins at acidic and alkaline pH. Condition 1 (**A**): pH 6.5, 0.5% enzyme, 1:8 SLR, and 45 °C for the EAEP$_1$ and pH 6.5, 1:8 SLR, and 45 °C for the AEP$_1$. Condition 2 (**B**): pH 9.0, 0.5% enzyme, 1:10 SLR, and 50 °C for the EAEP$_2$ and pH 9.0, 1:10 SLR, 50 °C for the AEP$_2$. Data were analyzed by a two-way ANOVA followed by Tukey's post-hoc test. Different letters indicate a significant difference between samples at $p < 0.05$.

EAEP$_1$ and AEP$_1$ skim proteins (Condition 1) had very similar solubilities at pH 5.0 (Figure 8A), suggesting no increment in protein solubility when using enzyme and extraction pH 6.5. However, the use of enzyme and alkaline pH (9.0) during the extraction (Condition 2) resulted in the production of a skim fraction with increased solubility at pH 5.0 (Figure 8B). EAEP$_2$ skim protein solubility was significantly higher than the solubility of AEP$_2$ skim proteins (45 vs. 23%), regardless of the extraction time evaluated. Higher solubility of EAEP$_2$ skim proteins at pH 5.0, where almond protein solubility is unfavorable [35], is likely related to the enzymatic release of smaller and more soluble peptides [5,34,38]. As shown in the SDS-PAGE gel, nearly complete hydrolysis of amandin, an insoluble protein under acidic pH, was observed in the EAEP skim for both conditions studied. The hydrolysis of amandin into smaller peptides could explain the higher solubility of the EAEP skim at acidic pH compared with the AEP skim. Our results are in agreement with the literature where the use of proteases resulted in the production of proteins with higher solubility at lower pH [5,16,34,38]. However, despite the higher extent of hydrolysis of EAEP$_1$ skim proteins (Condition 1), compared with EAEP$_2$ skim proteins (Condition 2), reduced solubility was observed for EAEP$_1$ skim proteins. Reduced solubility of EAEP$_1$ skim proteins is likely the result of excessive degree of hydrolysis which might have resulted in the exposure of hydrophobic groups initially buried in the core of the protein [35,39]. This trend, however, was not observed at pH 9.0, where almond protein solubility is generally favored [31]. AEP and EAEP skim protein solubilities were very similar at pH 9.0 for both extraction conditions (98% for Condition 1 vs. 99.5% Conditions 2, at 60 min of extraction time). Our results are in agreement with those reported by Souza et al. [16] in which significantly higher solubility of EAEP skim proteins extracted from almond cake was observed at pH 5.0, compared with the solubility of AEP skim proteins. Similar solubilities were also reported for AEP and EAEP skim proteins at pH 9.0.

## 4. Conclusions

The use of a fractional factorial design and subsequent validation of the effects of important reaction parameters on the extraction and separation of oil and protein from almond flour led to high extraction yields with a significant reduction of processing time. Although similar oil and protein extraction yields could be achieved at pH values ranging from 6.5–9.0 and SLR ranging from 1:8–1:10, the use of a more alkaline pH (9.0) resulted in significant reaction time reduction and improved separation and functionality of the extracted compounds. Experimental validation of best extraction conditions (pH 9.0, 0.5% of NP2M, 1:10 SLR, 50 °C, 60 min) resulted in 74.5% of oil and 72.3% of protein extraction. The use of enzyme to assist the extraction did not lead to significant increments in oil and protein extraction yields compared with the control (without enzyme), evidencing the major role of other processing parameters such as pH, SLR, and reaction time. However, the use of enzyme and alkaline pH (9.0) during the extraction resulted in the production of more soluble proteins for applications at acidic pH (5.0), compared when not using enzyme or using pH 6.5 during the extraction. At alkaline pH (9.0), high solubility was observed for both AEP and EAEP skim proteins (95%), regardless of the extraction conditions used. The higher concentration of soluble hydrolyzed peptides in the EAEP skim produced under alkaline extraction conditions could enable its use for specific food or feed applications involving acidic pH. However, it is important to highlight the increase in processing costs due to the use of enzyme to assist the extraction. Subsequent studies evaluating the use of reduced amounts of enzyme (<0.5%) during the extraction or enzyme recycling from the cream de-emulsification to the extraction are needed to minimize enzyme costs. In addition, the implications of the use of enzyme during the extraction on other functionality properties as well as on the resistance of the cream fraction produced by the EAEP process towards subsequent de-emulsification warrant further investigation.

**Author Contributions:** Conceptualization, J.M.L.N.d.M.B.; methodology, J.M.L.N.d.M.B., M.I.R., F.F.G.D. and N.M.d.A.; software, M.I.R.; validation, J.M.L.N.d.M.B., M.I.R., N.M.d.A.; formal analysis, J.M.L.N.d.M.B., N.M.d.A., M.I.R., and F.F.G.D.; resources, J.M.L.N.d.M.B.; writing-original draft preparation, N.M.d.A., F.F.G.D., J.M.L.N.d.M.B.; writing-review and editing, F.F.G.D., J.M.L.N.d.M.B., M.I.R.; supervision, J.M.L.N.d.M.B., project administration, J.M.L.N.d.M.B.; funding acquisition, J.M.L.N.d.M.B.

**Funding:** Funding for this project was made possible by the U.S. Department of Agriculture's (USDA) Agricultural Marketing Service through grant AM170100 XXXXG011. Its contents are solely the responsibility of the authors and do not necessarily represent the official views of the USDA.

**Acknowledgments:** This research was partially supported by a fellowship from the Coordination for the Improvement of Higher Education Personnel Program (CAPES) in Brazil.

**Conflicts of Interest:** The authors declare no conflict of interest.

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
