# Peer review of "Effects of Processing Conditions on the Simultaneous Extraction and Distribution of Oil and Protein from Almond Flour"

_processes, doi:10.3390/pr7110844_

Round 1
Reviewer 1 Report
line 107 oi- revise to oil Please remove the vertical lines (Tables 1-3). line 169 AOCS method revise to AOAC method line 360 De Moura et al. should check reference 11 or revise to Moura et al. line 671 reference 30. Please add journal name

Author Response
On behalf of all authors I am pleased to submit a revised version of the manuscript entitled “Effects of Processing Conditions on the Simultaneous Extraction and Distribution of Oil and Protein from Almond Flour”. We have modified the manuscript taking into account all thoughtful comments and questions raised by the reviewers. We are very grateful for all suggestions and comments, which resulted from the careful reading of this manuscript, as they helped to improve its quality.
Juliana Bell
Response to Reviewer #1 Comments
line 107 oi- revise to oil
-Corrected as requested.
Please remove the vertical lines (Tables 1-3).
-Vertical lines were removed as requested.
line 169 AOCS method revise to AOAC method
-Appropriate correction was made.
line 360 De Moura et al. should check reference 11 or revise to Moura et al. line 671 reference 30. Please add journal name
-The references were revised and modified as requested.
Reviewer 2 Report
This paper presents a very interesting study investigates the processing conditions of the simultaneous extraction of oil and proteins from almond flour. It focuses on EAEP (enzyme-assisted aqueous protein extraction).
The paper is well written, correctly illustrated, and the structure is adequate, but it is too long, and the results could be shortened. The introduction is informative and clearly states the objectives. Materials and methods are detailed, and the methodology seem rigorous and adequate. In particular, the preliminary comparison of two enzymes (proteases), and the systematic use of design of experiments or EAEP with a control (AEP), is particularly appreciated. The analysis not only includes the results of the extraction process in three phases (insoluble, skim and cream, respectively), but also the properties of the protein fraction, which is valuable. As already stated, the description of the results is too long, and their discussion could be reduced (see major points). Finally, the conclusion, as the abstract, clearly summarizes the key findings. So, the main weaknesses of the paper are, thus, that the final interest of EAEP in this process, and some of the statistical analyses of the results that must be revised.
As a conclusion, the opinion of the Reviewer is that this paper is original, that the experiments sound rigorously conducted, and that it could be accepted for publication after major revision. Major and minor points are appended below.
Major points
In section 3.2, the discussion of Table 2 and Table 3 is very long, while most of the p-values are higher than 0.05 (5%). This could probably be shortened. In section 3.2, a question that can result from the previous point is the range of study of the factors which is rather limited. While the effects of pH and SLR become statistically significant at 180 min, temperature between 45°C and 55°C, and enzyme content are never significant. However, these are key parameters on process cost. To decrease the cost of the process, one can regret that lower temperature and lower enzyme content were not investigated (except 0 in AEP). So, can the authors better justify their choice? Also, could it be interesting to investigate lower T and enzyme content? The conclusion is clear on the pros and cons of EAEP applied to almond flour in comparison to AEP without enzymes. A key question is, however, the price of the enzymes which increases the cost of the extraction process. Considering the advantages of EAEP (enhanced hydrolysis and solubility…) and the limited knowledge on other changes in protein functionality, can the use of EAEP be affordable? This point should be discussed at least in the conclusion.
Minor points
The abstract should be self-sufficient. Abbreviations, such as EAEP should be defined in the abstract. In the captions of the figures, the authors use lowercase letters (a), (b)…, whereas uppercase letters are shown in the figure. Please, correct the captions. The presentation of statistical results can be improved. For example, 70.6 +/- 3.6 % should be 71 +/- 4% because the second digit makes no sense in the confidence interval.
Author Response
On behalf of all authors I am pleased to submit a revised version of the manuscript entitled “Effects of Processing Conditions on the Simultaneous Extraction and Distribution of Oil and Protein from Almond Flour”. We have modified the manuscript taking into account all thoughtful comments and questions raised by the reviewers. We are very grateful for all suggestions and comments, which resulted from the careful reading of this manuscript, as they helped to improve its quality.
Juliana Bell
Response to Reviewer #2 Comments
2) This paper presents a very interesting study investigates the processing conditions of the simultaneous extraction of oil and proteins from almond flour. It focuses on EAEP (enzyme-assisted aqueous protein extraction).
The paper is well written, correctly illustrated, and the structure is adequate, but it is too long, and the results could be shortened. The introduction is informative and clearly states the objectives. Materials and methods are detailed, and the methodology seem rigorous and adequate. In particular, the preliminary comparison of two enzymes (proteases), and the systematic use of design of experiments or EAEP with a control (AEP), is particularly appreciated. The analysis not only includes the results of the extraction process in three phases (insoluble, skim and cream, respectively), but also the properties of the protein fraction, which is valuable. As already stated, the description of the results is too long, and their discussion could be reduced (see major points). Finally, the conclusion, as the abstract, clearly summarizes the key findings. So, the main weaknesses of the paper are, thus, that the final interest of EAEP in this process, and some of the statistical analyses of the results that must be revised.
As a conclusion, the opinion of the Reviewer is that this paper is original, that the experiments sound rigorously conducted, and that it could be accepted for publication after major revision. Major and minor points are appended below.
We are thankful for the suggestions and we have addressed the reviewer’s comments throughout the manuscript as detailed below.
Major points
1)In section 3.2, the discussion of Table 2 and Table 3 is very long, while most of the p-values are higher than 0.05 (5%). This could probably be shortened.
We agree with the reviewer and we have shorted the discussion to make this section more concise. A couple of sentences and repetitive explanations were deleted.
2)In section 3.2, a question that can result from the previous point is the range of study of the factors which is rather limited. While the effects of pH and SLR become statistically significant at 180 min, temperature between 45°C and 55°C, and enzyme content are never significant. However, these are key parameters on process cost. To decrease the cost of the process, one can regret that lower temperature and lower enzyme content were not investigated (except 0 in AEP). So, can the authors better justify their choice? Also, could it be interesting to investigate lower T and enzyme content? Further study could be conducted with lower amount of enzyme? The conclusion is clear on the pros and cons of EAEP applied to almond flour in comparison to AEP without enzymes. A key question is, however, the price of the enzymes which increases the cost of the extraction process. Considering the advantages of EAEP (enhanced hydrolysis and solubility…) and the limited knowledge on other changes in protein functionality, can the use of EAEP be affordable? This point should be discussed at least in the conclusion.
We absolutely agree with the reviewer comment. Our choice of evaluating the 45-55oC range was based on the literature, which shows evidence that temperature around 50oC often favors overall extractability, and on based on the best temperature for the enzyme used in our study (50 oC) (Rui et al. 2009 _Journal of Food Engineering, 93, 482-486, 2009 and Sharma et al. 2002_ Journal of the American Oil Chemists’ Society, 79(3), 215-218, 2002). Although the use of enzyme concentration around 0.5% has shown to favor extractability (Rosenthal, et al., Enzyme and Microbial Technology 28, 499–509, 2001), the investigation of reduced amount of enzyme (<0.5%) as well as the evaluation of strategies to recycle the enzyme from the cream de-emulsification to the extracted are needed to minimize enzyme costs. We have added a comment in conclusions to highlight this point. A brief discussion was added to the conclusion as suggested (lines 587-590).
Minor points
3)The abstract should be self-sufficient. Abbreviations, such as EAEP should be defined in the abstract.
The definition of the abbreviation was added to the abstract as suggested.
4)In the captions of the figures, the authors use lowercase letters (a), (b)…, whereas uppercase letters are shown in the figure. Please, correct the captions.
The letters were standardized throughout the text as requested.
5)The presentation of statistical results can be improved. For example, 70.6 +/- 3.6 % should be 71 +/- 4% because the second digit makes no sense in the confidence interval.
The results were modified in the text, which now shows only one digit.
Reviewer 3 Report
The article by Almeida et al. provides a novel and thorough statistical analysis and study on the effects of various processing conditions such as pH, temperature, solids-to-liquid ratio, amount of enzyme on the yields and distribution of oil and protein in various fractions of the process. The article also focuses on the extraction in alkaline conditions and possible applications of EAEP in acidic pH. The authors have used a detailed and sophisticated experimental design which considers all processing variables and which reduces the number of experiments thereby improving the process efficiency and productivity. I suggest the manuscript should be considered for publication with minor changes. My comments are as follows:
There are a few grammatical errors in the manuscript. I suggest the authors should run a spell-check and proofread the manuscript to remove those errors. Figure 2: Line no. 232: TOP: Total oil extraction. Please change TOP to TOE. Line no. 281-283. “Overall, free oil yield decreased….
I suggest the authors should provide a little more explanation on the change in free oil yield with change in the extraction time and in turn the change in amount of oil in cream fraction.
Line no. 359: …Significant effect (-0.58) (p<0.05) at 40 min (Table 2). Please change “Table 2” to “Table 3”. Page no. 14; Line 429, 430: It is mentioned that the use of enzyme did not play a key role in improving oil extraction yields. However, the total oil extraction yield has decreased significantly for 20 mins from EAEP to AEP. A brief explanation on this would be useful for the reader to understand the trend and governing parameters better.
Author Response
On behalf of all authors I am pleased to submit a revised version of the manuscript entitled “Effects of Processing Conditions on the Simultaneous Extraction and Distribution of Oil and Protein from Almond Flour”. We have modified the manuscript taking into account all thoughtful comments and questions raised by the reviewers. We are very grateful for all suggestions and comments, which resulted from the careful reading of this manuscript, as they helped to improve its quality.
Juliana Bell
Response to Reviewer #3 Comments
3) The article by Almeida et al. provides a novel and thorough statistical analysis and study on the effects of various processing conditions such as pH, temperature, solids-to-liquid ratio, amount of enzyme on the yields and distribution of oil and protein in various fractions of the process. The article also focuses on the extraction in alkaline conditions and possible applications of EAEP in acidic pH. The authors have used a detailed and sophisticated experimental design which considers all processing variables and which reduces the number of experiments thereby improving the process efficiency and productivity. I suggest the manuscript should be considered for publication with minor changes. My comments are as follows:
1)There are a few grammatical errors in the manuscript. I suggest the authors should run a spell-check and proofread the manuscript to remove those errors.
The manuscript was revised, and a spell check was performed as suggested.
2)Figure 2: Line no. 232: TOP: Total oil extraction. Please change TOP to TOE.
Corrected as requested.
3) Line no. 281-283. “Overall, free oil yield decreased….I suggest the authors should provide a little more explanation on the change in free oil yield with change in the extraction time and in turn the change in amount of oil in cream fraction.
A paragraph was added explaining this change as suggested (lines 284-286)
4)Line no. 359: …Significant effect (-0.58) (p<0.05) at 40 min (Table 2). Please change “Table 2” to “Table 3”.
Corrected as requested.
5)Page no. 14; Line 429, 430: It is mentioned that the use of enzyme did not play a key role in improving oil extraction yields. However, the total oil extraction yield has decreased significantly for 20 mins from EAEP to AEP. A brief explanation on this would be useful for the reader to understand the trend and governing parameters better.
Although there is small increment in TOE for certain EAEP conditions compared with AEP conditions if performing a one -way Anova within the same extraction condition (Treatment 1 or Treatment 2), most of these effects become not statistically significant when comparing both treatments by a two-way Anova at different reaction times, indicating the key role of pH to improve oil extractability within the conditions tested. While TOE at EAEP2 (75%, pH 9.0 and enzyme use) is significantly higher than TOE at EAEP1 (61%, pH 6.5 and enzyme) and AEP1 (58%, pH 6.5, no enzyme), similar TOE can be achieved at AEP2 (70%, pH 9.0, no enzyme), evidencing the pH effect on improving oil extractability.